# Virus-Induced Gene Silencing (VIGS): A Powerful Tool for Crop Improvement and Its Advancement towards Epigenetics

**DOI:** 10.3390/ijms24065608

**Published:** 2023-03-15

**Authors:** Sumer Zulfiqar, Muhammad Awais Farooq, Tiantian Zhao, PeiPei Wang, Javaria Tabusam, Yanhua Wang, Shuxin Xuan, Jianjun Zhao, Xueping Chen, Shuxing Shen, Aixia Gu

**Affiliations:** State Key Laboratory of North China Crop Improvement and Regulation, Key Laboratory of Vegetable Germplasm Innovation and Utilization of Hebei, Collaborative Innovation Centre of Vegetable Industry in Hebei, College of Horticulture, Hebei Agricultural University, Baoding 071000, China

**Keywords:** VIGS heritable epigenetics, VIGS vectors, reverse genetics, plant RNA, DNA viruses, biotic and abiotic stresses

## Abstract

Virus-induced gene silencing (VIGS) is an RNA-mediated reverse genetics technology that has evolved into an indispensable approach for analyzing the function of genes. It downregulates endogenous genes by utilizing the posttranscriptional gene silencing (PTGS) machinery of plants to prevent systemic viral infections. Based on recent advances, VIGS can now be used as a high-throughput tool that induces heritable epigenetic modifications in plants through the viral genome by transiently knocking down targeted gene expression. As a result of the progression of DNA methylation induced by VIGS, new stable genotypes with desired traits are being developed in plants. In plants, RNA-directed DNA methylation (RdDM) is a mechanism where epigenetic modifiers are guided to target loci by small RNAs, which play a major role in the silencing of the target gene. In this review, we described the molecular mechanisms of DNA and RNA-based viral vectors and the knowledge obtained through altering the genes in the studied plants that are not usually accessible to transgenic techniques. We showed how VIGS-induced gene silencing can be used to characterize transgenerational gene function(s) and altered epigenetic marks, which can improve future plant breeding programs.

## 1. Introduction

Virus-induced gene silencing (VIGS) is an effective method to silence the gene that uses a plant’s antiviral defensive mechanism to suppress the expression of specific invasive viral transcripts [1]. For the first time, van Kammen coined the term VIGS to characterize the phenomenon of ‘recovery from viral infection’ [2]. The first VIGS vector was constructed using the tobacco mosaic virus (TMV) by Kumagai et al. (1995), which efficiently silenced the *NbPDS* gene expression by inoculating in vitro RNA transcripts into *Nicotiana benthamiana*. Consequently, the plants produced had an albino phenotype [3]. Since then, the term ‘VIGS’ has meant any approach that uses recombinant viruses to inhibit endogenous gene expression [4,5]. This transcription suppression technique has been mainly applied in the field and on horticultural crops [1], and is now also being applied in forest trees [6], i.e., *Populus euphratica*, *Populus canescens* [7], *Hevea brasiliensis* [8], and *Olea europaea* [9,10]. VIGS is an RNA-mediated reverse genetics technique in eukaryotes that involves different classes of small RNAs with specialized roles in gene silencing [11]. These RNAs have been used to identify gene functions in plant species at an individual or at a large scale in a high-throughput fashion [12]. VIGS is effective in trans-generationally inherited epigenetic modifications [13]. 

Non-genetic variations, or “epigenetic diversity”, can influence plant phenotypes [14]. Studying heritable gene expression and heritable epigenetic marks that cause heritable phenotypic diversity [15], influencing fitness and so being subject to natural selection, is known as epigenetics. The VIGS functional genomics approach consists of three primary steps: first, creating viral vectors containing targeted genes that need to be silenced [16]. Second, the inoculation of the viral vectors into the selected plant host [17]. Third, the silencing of the targeted gene in the plant as a defense mechanism against the viral vectors which are introduced into the host plants [18]. 

The process of VIGS occurs in the cytoplasm of the cell and is regarded as Posttranscriptional gene silencing (PTGS) in plants, quelling in fungi, and RNAi in animals [19,20]. PTGS is an epigenetic phenomenon associated with methylation of the coding sequence [21], which results in the sequence-specific degradation of endogenous mRNAs [22,23]. It is induced by the delivery of targeted genes into a viral vector to reduce the expression of a gene of interest to characterize the sequence-specific phenotype [12,22,24,25,26]. PTGS occurring due to the DNA methylation of the genes eliminates the assumption that PTGS-linked DNA methylation signals can move trans-generationally without PTGS signals [21]. Epigenetic marks and the heritability of virus-induced silencing are linked to the methylation of DNA. Plants are inoculated with a viral vector (DNA or RNA) that carries a sequence corresponding to the targeted gene [27,28,29]. The inoculation leads to the activation of endogenous RNA-directed RNA polymerase (RDRP), which replicates and produces viral dsRNA [30]. These dsRNAs are recognized by the Dicer enzyme analog (also called the mother of proteins), which cleaves them into siRNAs duplexes that are approximately 21–24 nt in length [31,32,33,34]. In cells, RNA-dependent RNase amplifies siRNAs, which combine with AGO protein-containing effector complexes [35] to form the RNA-induced silencing complex (RISC) [36]. AGO enables siRNA to attach to a specific RNA or DNA target through a sequence-homology-dependent mechanism. RISC uses these siRNAs to specifically interact with homologous RNA in the cell, thereby leading to endo-nucleolytic cleavage and the translational inhibition of the cognate target mRNA, causing PTGS (Figure 1) [12,37]. According to studies, the secondary siRNAs appear to improve VIGS maintenance and dissemination, which are produced by the cleavage of dsRNA synthesized by the host RDRP using the primary siRNA as a template [38,39]. Simultaneously, the AGO complex interacts with target DNA molecules in the nucleus, causing transcriptional repression via DNA methylation at the 5′ untranslated region (5′UTR), which results in transcriptional gene silencing (TGS) [37,40,41,42]. On the other hand, the mechanism of virus-induced epigenetic modification for gene silencing begins with generating siRNA through Dicer, which targets the chromatin-bound scaffold RNA in association with the AGO portion. The transcription of scaffold RNA occurs mainly through plant-specific PolV, or, in some instances, RNA polymerase II also mediates this process [43]. DNA methylation is a prerequisite for Pol V recruitment [44]; for this purpose, DNA methyltransferase reaches the chromatin locus to introduce methyl groups on C residues at CG, CHG, and CHH contexts. These methyl groups can result in heritable gene silencing if they are in proximity to promoter sequences [45]. Heritable epigenetic modifications can alter the phenotypic variations that can trigger natural selection and play a significant role in the evolutionary process of species [15]. In this review, we discuss novel insights and applications of VIGS for its molecular mechanism, biotic and abiotic stress tolerance gene identified via VIGS for crop improvement, heritable epigenomic modifications, gene functional analysis, the development of a virus-induced base-editing technique, and the VIGS application constraints that remain to be overcome despite its immense potential. To this date, there are no published review articles that demonstrate the mechanism behind VIGS-induces heritable epigenetic modification in plants. In summary, VIGS can play a major role in understanding molecular mechanisms, which will have a direct impact on developing crop varieties with better agronomic traits and stress tolerance. 

## 2. VIGS-Induces Heritable Epigenetics Modification in Plants

To achieve epigenetic silencing, the viral vector insert must correspond to the transgene promoter rather than to the coding sequence [15]. Early DNA methylation is an epigenetic mark that is subsequently reinforced via the PolIV pathway of RNA-directed DNA methylation (RdDM), leading to a heritable epigenome. DNA methylation causes genetically inherited and/or transient alterations in chromatin structure and gene expression that do not make a significant difference in nucleotide sequence, and leads to genomic imprinting and gene silencing [46]. PolV has also been linked to the production of scaffold RNAs in *FLOWERING WAGENINGEN* (FWA) gene promoters, which are binding sites of AGO-bound sRNAs that direct DNA methylation to the adjacent chromatin [47] (Figure 2). A mutation in PolV results in the complete loss of VIGS-RdDM, as this protein is involved in both the formation and maintenance of silencing via RdDM. 

Furthermore, RNA-independent maintenance is dependent on the DNA methyltransferases MET1 and CMT3 recognizing hemimethylated Cs in the symmetrical context of freshly duplicated DNA motifs for the maintenance of epigenetic marks. However, the RNA-dependent maintenance is sequence motif independent as it involves canonical PolIV-RdDM, in which the 24-nt sRNA biogenesis proteins produced through DCL3 proteins are recruited to the genomic location of primary RdDM by methyl DNA-binding proteins (Figure 3). The DNA methyltransferases would then be directed to the unmethylated strand of newly replicated DNA by the 24-nt sRNAs. To create gene silencing, the target locus *dcl2/4* should have a mutation and a functional DCL3 to provide maximum reinforcement through the RNA-dependent maintenance mechanism [47,48]. A target containing a high percentage of C residues in the CG context will ensure RNA-independent maintenance efficiency, which further improves the system. For species in which the epigenetic maintenance mechanism is active, VIGS has been strengthened. 

There are several ways to induce DNA methylation in plants artificially: siRNA-mediated DNA methylation (VIGS) [47], inverted repeat transgenes [49,50], using programmable DNA-binding proteins to directly target methylation (zinc finger proteins) [44,45], CRISPR-dCas9 [51], and virus-induced transcriptional gene silencing-mediated DNA methylation (ViTGS-mediated DNA methylation) [39]. With the advancement of DNA methylation induced by VIGS, new stable genotypes of desired traits are being developed in plants [52]. To describe this novel process, which involves the de novo formation of heritable epigenetic marks in plants, Bond et al. (2015) used VIGS in wild-type and mutant *Arabidopsis,* in which they showed that TRV:FWAtr infection leads to transgenerational epigenetic silencing of the FWA promoter sequence [47]. Fei et al. (2021) illustrate that ViTGS-mediated DNA methylation is fully established in the parental lines and will be passed down to succeeding generations [39]. Fei et al. (2021) demonstrate that sRNA-induced transgenerational epigenetic gene modifications are possible and give definite evidence that 100% sequence complementarity between the target DNA sequence and the sRNAs is not required for transgenerational RdDM [13]. This brings new opportunities to study the close relationship between the epigenome and invading molecules like transposons and viruses. Such epigenetic gene silencing appears unaffected and durable over numerous generations. By using this type of genotyping, epigenetic VIGS could be employed in breeding programs in the future to better understand the structure and function of genes. Previously, only a limited amount of genetic and molecular evidence supported transgenerational epigenetic inheritance processes. A symmetry change in the flowers of *Linaria vulgaris* (toadflax) is a well-known plant example of transgenerational epigenetic inheritance [53]. Somatic reversion to bilateral flower symmetry coincides with decreased DNA methylation, whereas a change from bilateral to radial symmetry has been linked to increased DNA methylation in the upstream promoter area of the *Lcyc* locus [54]. Despite the fact that many scientists have speculated on the potential use of epigenetics in breeding [55,56,57,58], there are still many loopholes that need to be plugged for this new science to become a mainstream plant breeding technique, including its inherent instability, the heterogeneity of epigenetic programming, and the challenges in determining its precise treatment. 

## 3. Development of Virus-Induced Base-Editing Technique

Precise base-editing technologies have enormous potential for quickening crop development and facilitating research into plant gene function [59,60]. However, to date, it has proven challenging to accomplish heritable base editing in many plants, including *Arabidopsis*. According to several studies [61,62], transgenic *Arabidopsis* plants expressing base-editing reagents frequently exhibit somatic mosaicism in the first generation, and multiple generations are required to fix edited alleles [60,62]. Recently, using RNA or DNA viruses, heritable targeted mutagenesis has been accomplished [63,64,65].

Liu et al. (2022) developed a method to produce heritable base-editing by using RNA viral vectors to deliver sgRNAs in *Arabidopsis* mutant lines expressing a cytidine deaminase base-editor, displaying loss-of-function mutations at *PDS3 (PHYTOENE DESATURASE 3).* These mutant lines showed a photobleached phenotype because of carotenoid biosynthesis inhibition, and gain-of-function mutations at *CESA3* (*CELLULOSE SYNTHASE 3)* that create a S983F amino acid substitution result in conferring tolerance to the cellulose biosynthesis-inhibiting chemical compound C17 (5-(4-chlorophenyl)-7-(2-methoxyphenyl)-1,5,6,7-tetrahydro-[1,2,4] triazolo [1,5-a] pyrimidine). This method makes it possible to conduct high-throughput gene function analyses on plants from the M0 generation. They showed that homozygous mutant plants could be recovered in the M1 progeny. *Arabidopsis, Nicotiana benthamiana*, and tomato (*Solanum lycopersicum*) [66] have also exhibited heritable mutations as a result of this viral delivery technique using TRV, and it is likely that this technique can be applied to other plant species. In fact, recent research using barley stripe mosaic virus vectors showed that wheat (*Triticum aestivum*) infection can cause heritable editing (Table 1) [67]. This virus-induced base-editing technique will provide new opportunities for advancing functional genomics and crop enhancement because of its simplicity, dependability, and adaptability.

## 4. Basis of Insert Carrier Selection and the Necessity of Developing New Carriers

To date, more than 50 VIGS vectors [71] have been created by modifying plant DNA and RNA viruses for either dicots or monocots, or both [72] (Appendix A), and these vectors have been extensively employed to identify gene functions in plant defense response pathways, symbiosis, nematode resistance, nutrient acquisition, abiotic stress response, metabolic pathways, and cellular functions [33,73,74]. Four of them, including the potato virus X (PVX), tobacco rattle virus (TRV), cucumber mosaic virus (CMV), and apple latent spherical virus (ALSV), have been demonstrated to cause heritable TGS [47,75,76,77]. The development of the VIGS vector is dependent on the kind of species and infection efficiency. For instance, single-stranded RNA viruses are widely used viral carriers for the establishment of VIGS system due to their low molecular weight and high infection penetrance. They have been successfully used in Poaceae crops through barley stripe mosaic virus (BSMV) [78,79] and in a wide variety of dicots through the tobacco rattle virus (TRV). Single-stranded DNA viruses (ssDNA) are peculiar to a specific host due to their large genome structure and limited mobility. They have been successfully employed in *N. benthamiana* and *Manihot esculenta* (cassava) through the tomato mottle virus (ToMoV) and the African cassava mosaic virus (ACMV) [1].

## 5. VIGS-Vectors for Gene Functional Analysis

### 5.1. RNA Based-VIGS Vectors

Previously, the majority of the VIGS vectors used for gene silencing were RNA-based viruses because RNA viruses have been shown to induce silencing in a variety of host plants, including *Capsicum frutescens* (chili pepper), *Capsicum* (bell pepper), *Arabidopsis thaliana* (thale cress)*, Nicotiana benthamiana* (benth or benthi)*,* cotton plants, *Solanacae*, and several other monocotyledon plants, such as *Oryza sativa* (rice), *Hordeum vulgare* (barley), and *Zea mays* (maize), to eliminate endogenous transcripts [80]. When a positive single-stranded RNA (+ssRNA) virus enters a plant cell, it is translated by the virally encoded replicase (with methyltransferase, helicase, and virus-encoded RNA-dependent RNA polymerase (RdRP activity)) at the cellular membrane, and then the virus’s replication occurs at the internal invaginations of the membrane mediated by more than 100 proteins, including translation, initiation, and other elongation factors [81]. Then, replicase generates a negative-strand RNA (-ssRNA) virus complementary to the viral genome. At the 3` end of a viral genome, a specific RNA structure is present, to which replicase attaches and initiates RNA synthesis. The viral genome and its negative-strand complementary sequence are then copied many times by the replicase. [82]. The resultant viral ssRNAs are mostly secondary or tertiary structures, which are the most dominant kind of viral short-interfering RNA (vsRNA) [29]. It is important to mention that a database for VIGS vectors and phenotype genomes for *N. benthamiana* has been established, facilitating functional genomics and phenomics in this species, and perhaps in other Solanaceae as well [83].

To test the effect of the RNA-based VIGS vector on other families, such as Fabaceae (pea and soybean), the pea early-browning virus (PEBV), and bean pod mottle viruses (BPMV), were utilized [84,85,86]. Several genes, including *pds* (involved in the carotenoid biosynthesis pathway), *uni* (homolog of the *flo* and *lfy* genes from *Arabidopsis*)*,* and *Korrigan1* (involved in cellulose biosynthesis), were silenced using PEBV [86]. In the case of *pds*, the silenced plant showed photo-bleached patches within ten days of inoculation, and more than half of *uni* silenced plants had tilted leaf growth. Similarly, *korrigan1*-silenced plants showed an extreme dwarf phenotype. Multiple genes were silenced at the same time, although the effects were minor when compared to plants in which the genes were silenced individually [20].

A cucumber mosaic virus (CMV)-derived VIGS vector was developed to silence chalcone synthase (*chs*) in *A. thaliana* [77], *Antirrhinum majus* (snapdragon) [87], *N. benthamiana* [88], and *sf3h1* along with *chs* in soybean, and silenced plants indicated decreased pigmentation in the seed coat and lower flavonol level in the whole plant [89].

VIGS has also been applied to several monocotyledon plants. One of the latest VIGS vectors for monocot plants, the foxtail mosaic virus (FoMV), has been used to effectively silence PDS in maize [90]. The barley stripe mosaic virus (BSMV) has been successfully applied to silence the gene in species belonging to the Triticeae genus, for example, *Triticum aestivum* (wheat), *Avena sativa* (oats), and *Secale cereal* (rye) [91,92,93].

Now, RNA-based vectors have also been used to induce heritable gene editing. Recently, Ma et al. (2020) developed vectors based on the sonchus yellow net nucleorhabdovirus (SYNV), a negative-sense RNA virus [94]. When SYNV enters the germline, as described by Ellison et al. (2020), heritable mutations can be recovered solely through infection rather than tissue culture and transgenesis. The virus’s ability to penetrate into the meristematic region will determine heritable editing in plants using viral vectors [63]. Unlike most plant viruses, TRV can transiently assault the meristematic region in the earlier stages of the infection cycle. In accordance with this, TRV has been effectively utilized to silence meristem genes such as *NFL* (an ortholog of *Arabidopsis LEAFY* [95]) as well as floral homeotic genes such as *DEFICIENS* (an ortholog of *Arabidopsis AP3* [96]) in *N. benthamiana.* The addition of tRNA^Ileu^ (tRNA isoleucine) to sgRNAs in TRV promotes systemic movement and is capable of inducing effective somatic and heritable editing in target genes in *Arabidopsis* and *N. benthamiana* [63,65].

### 5.2. DNA Based-VIGS Vectors

Since the initial discoveries more than 25 years ago, numerous plant DNA viruses have been sequenced, cloned, and modified to deliver exogenous sequences into plant cells. Delivering sequences encoding components for the major classes of site-specific nucleases (SSNs) has recently proved the efficacy of DNA viral vectors for gene editing in a diverse variety of host species [97]. The vast majority of both foundational and ongoing works in viral engineering describe work with Geminiviridae, single-stranded DNA (ssDNA) viruses that comprise the largest known family of DNA viruses in plants [98]. Other DNA-based viruses, such as ACMV and cotton leaf crumple virus (CLCrV), have been effectively transformed into VIGS vectors and successfully used in cassava and cotton plants [99]. Silenced plants revealed white and yellow blotches on the leaves after the biolistic delivery of the ACMV-VIGS vector to silent the magnesium chelates (*su*) responsible for the chlorophyll production enzyme [100].

The ssDNA, upon entering the cell, undergoes rolling circle replication (RCR) using the host cell proteins [82]. The replication process occurs close to the nucleus, where the viral DNA aggregates as circular double-stranded DNA (dsDNA) mini-chromosomes of the cell cycle [29]. Hanley-Bowdoin et al. (2013) reported that gemniviruses can replicate in the G phase by exercising their control on the host cell replication machinery while the host cell is in a normal non-replication state [82]. The host cell RNA polymerase II (Pol II) transcribes the dsDNA genome, which contains the origins of replication and promoter elements and is often bidirectional. As a result, bidirectional synthesized transcripts are proposed as dsRNA precursors of primary viral short interfering RNAs [29].

VIGS has also been successfully introduced into additional dicot species, such as *Brassica napus*, using the cabbage leaf curl virus (CaLCuV) as a vector. Sense sequences were inserted into the CaLCuV-A component and transformed into seedlings with the CaLCuV-B component, utilizing biolistic delivery to downregulate targeted genes [101]. In *Brassica rapa,* through the particle bombardment of plasmid DNA, turnip yellow mosaic virus (TYMV) effectively silenced the *BrPDS* gene [102]. Another DNA-based vector, the rice tungro bacilliform virus (RTBV), is a recently constructed VIGS vector that was first introduced into rice; after that, it got expanded into other species as well, like *Cynodon dactylon* (Bermuda grass), and *Zoysia japonica* (Korean lawn grass) [70,103]. DNA-based satellite viruses like the tomato yellow leaf curl China virus (TYLCCNV beta satellite), DNAβ tobacco curly shoot virus (TbCSV) alpha satellite (DNA1) and DNAβ bhendi yellow vein mosaic virus (BYMV) have also been modified as VIGS vectors for use in *N. benthamiana* [104].

Plant DNA viruses do not code for replicases, and vsRNA synthesis also does not require the host’s RNA Pol II for the transcription of their genome, as indicated in the cauliflower mosaic virus (CaMV) [105]. DNA virus-derived dsRNAs are generated from the annealing of the complementary sense-antisense strands and structured transcripts [29], and consequently, target gene fragments require accurate positioning where converging sense-antisense transcripts can bind to each other after the transcription of VIGS vectors.

### 5.3. Biotic and Abiotic Stress Tolerance and Their Potential for Crops Improvement via VIGS

Biotic and abiotic stresses are major environmental threats that significantly lower crop productivity. Examples of biotic stresses include a variety of living organisms, such as fungi, insects, nematodes, bacteria, and viruses, and abiotic stresses include drought, salt, cold, and heat [106,107]. Although plants are constantly exposed to various stresses that result in complex response interactions, agricultural output is nonetheless negatively affected. For instance, it has been reported that abiotic stresses can result in yield losses of more than 50% [108], while biotic stresses are thought to result in yield losses of about 35% [109]. There is an urgent need to better understand the complex responses of plants to single and combination stresses in order to ultimately increase crop tolerance to changing climatic circumstances. This is especially true in view of future climate change scenarios and severe weather conditions. As a result, plant breeders have adapted molecular genetics techniques to develop effective resistance in crop plants in less time. Plant viruses have contributed to plant genomic studies for decades [110]. The traditional VIGS systems have been successfully used to study plant responses to both biotic and abiotic stresses [111]. For example, TRV-VIGS has been utilized for the functional genomic analysis of more and more plant species, including most dicotyledonous species, some monocotyledonous plants, and even some trees. The gene function of many genes in different plant tissues and organs, from seeds, roots, stems, leaves, flowers, to fruits, has been revealed by TRV-VIGS in the past 20 years, these functions being related to plant growth and development, metabolic pathways, and the response to biotic and abiotic stress. Singh et al. (2017) recently used VIGS by using the cauliflower mosaic virus (CaMV) to demonstrate that the silencing of the *WsWRKY1* gene has a negative effect on the expression of defense genes, resulting in reduced tolerance to biotic stress. *WsWRKY1* is important as a metabolic engineering tool for simultaneously enhancing triterpenoid biosynthesis and plant defense due to its positive regulatory involvement in phytosterol and withanolides biosynthesis and its protection against biotic stress (bacterial and fungal infections) [112]. Additionally, Boevink et al. (2016) used VIGS to determine the mode of action of a novel fungal effector molecule in *Phytophtora infestans* (the pathogen that causes late blight disease in the Solanaceae family). This molecule promotes *P. infestans* infection by reducing the expression of common pathogenresponse genes like the jasmonic and salicylic acid response genes [113]. VIGS was also employed in recent studies to determine the link between auxin biosynthesis in tomato, which is controlled by cytosolic calcium signaling, and plant defense responses involving the phytosulfokine signaling molecule [114]. VIGS was also used to functionally characterize the tomato receptor-like kinase SILYK10, which is a component of the signaling pathway that permits the mycorrhizal fungus to access the plant’s roots [115]. In a recent study, VIGS was also employed to shed light on the relationship between auxin biosynthesis in tomatoes, which is controlled by cytosolic calcium signaling and plant defense responses involving the phytosulfokine signaling molecule [114].

## 6. VIGS’s Limitations and Its Potential Solutions

VIGS is now widely regarded as one of the most promising and effective tools for studying the function of many genes. Its key benefit is the ability to generate fast phenotypes without needing stable plant transformation. Compared to other tools such as T-DNA, chemical and physical mutagenesis, transposon insertion techniques, and other functional genome editing approaches such as CRISPR-Cas involved in silencing, VIGS is inexpensive [12]. Despite the improvements in VIGS protocol, there are still several problems that can limit its utility.

First, the virus vector can disrupt plant metabolism [116,117], affecting plant–microbe interaction. The insertion of a gene into the VIGS vector can prevent the virus from multiplying, and several viruses have been known to delete the inserted gene during multiplication and spread [118]. Furthermore, most viral vectors fail to produce silencing in meristematic tissue, resulting in the insufficient silencing of a target gene [64]. As a result, VIGS is increasingly being utilized to investigate gene functions [119] related to plant development that would otherwise be fatal and difficult to research in mutant plants. Moreover, the genotype of plant species can influence the VIGS construct’s effectiveness. Thus, for some plant species, a VIGS technique must be customized to each genotype [36,120].

The lack of an efficient delivery method for VIGS in plants is one of the major factors that can reduce the efficacy of VIGS; in dicots, the most widely used delivery method is *Agrobacterium*-mediated agro-infiltration. However, certain plant species are non-compatible with *Agrobacterium*-mediated transformation [18,121]. This can be addressed by introducing the VIGS vector through Rub inoculation with RNA transcripts or DNA bombardment. In addition, uneven or localized VIGS results in a lack of silencing in specific tissues, leading to ineffective virus movement. This can be resolved by providing a conducive environment for dissemination and by selecting an appropriate virus vector that can spread systemically throughout the host plant without removing the insert [25].

Furthermore, if the target sequence is similar to the hairpin sequence, silencing efficiency will be considerably improved [99,122]. Antisense RNA triggers are more efficient than sense RNA ones but less efficient than those in hairpin constructions. The production of hairpin RNAs is necessary for the onset of potent host gene silencing.

Off-target silencing in VIGS can happen due to the partial homology between siRNA and the unintended mRNA sequence. For example, in wheat leaves, smaller inserts (100 nt) were substantially more stable than bigger inserts (150–252 nt). They resulted in the persistent and considerable silencing of the target gene, mRNA [120]. Moreover, it has been reported that there is a negative association between insert length and insert stability and/or silencing efficiency for a VIGS system [123], such as PVX-VIGS in *N. benthamiana* [124], BSMV-VIGS in *Hordeum vulgare* [125], and CWMV-VIGS (Chinese wheat mosaic virus) in *Triticum aestivum* [123]. The likelihood of silencing is reduced when the siRNA and target mRNA have a nucleotide identity of less than 11 bp [126]. Generally, a fragment from the UTR region 3′- or 5′-is a good choice because it is usually more variable than the CDS and minimizes the risk of off-target silencing. On the other hand, when a high level of functional redundancy is expected across gene family members, VIGS constructs should be developed from conserved gene sections to target many or even all gene family members at the same time [127]. For this purpose, selecting an insert gene sequence using publicly available software can reduce the off-target silencing problem [128]. Previous studies have effectively used the ‘siRNA-scan’ software (http://bioinfo2.noble.org/RNAiScan/RNAiScan.htm (accessed on 5 January 2023 ) to select the potential target and prevent off-target effects using integrated databases [128].

Researchers have developed various VIGS inoculation methods that affect the viability of VIGS to better understand how genes work at different phases of plant growth. The infection solution’s concentration in agroinoculation procedures significantly impacts the gene-silencing efficacy of VIGS experiments. The *Agrobacterium* inoculum should be used fresh for each use, and if the concentration of the resuspension is low, it will work well, but if the solution concentration is higher than 1.0 OD_600_, it may cause necrosis on the infiltrated *N*. *benthamiana* leaves [129]. Moreover, in tomatoes, the Agrobacterium culture at OD_600_ = 1.0 resulted in efficient *SIPDS* gene silencing [130]. In contrast, an *Agrobacterium* concentration of OD_600_ =1.5 worked better for tomatoes [129]. Similarly, the concentration of *Agrobacterium* cultures for cotton seeds at OD_600_ = 1.5 for 90 min resulted in the efficient silencing of SSA-VIGS (seed soak agroinoculation VIGS) [131]. For the vacuum infiltration method, it was found that the optimal conditions included the vacuum treatment of an *Agrobacterium* concentration at OD_600_ = 0.3 for 30–60 s and then the co-cultivation of this with the same resuspension concentration for 15 h [132]. TA Cloning has been widely applied in genetic engineering compared to gene gun technology in the co-transformation of genes, for example, *ZxZF*, and *AREB* for abiotic stress tolerance in *Populus euramericana* by the particle bombardment method. Although gene gun technology is expensive, it can be used effectively in certain situations, with clear advantages [133]. Furthermore, the exogenous genes are expressed in an unreliable manner and can be quickly lost when transformed through bombardment. Consequently, there will be no transformants or chimeras, and the cell’s normal gene expression will be disrupted as well, which can potentially lead to co-inhibition [133].

Environmental factors can also affect the silencing of genes in plants, i.e., at high temperatures, most viruses lose their potency, resulting in a drop in virus concentration [36]. In *N. benthamiana*, a higher temperature promotes antiviral defense via RDRP6, and it results in a large number of small mRNAs [134] which will reduce the PTGS by preventing viral spread throughout the plant. A lower temperature results in higher viral concentrations and silencing [135]; TRV-VIGS, for example, increased silencing in petunia at the lower temperatures of 20 °C Day/18 °C night [136]. To meet the low temperature requirements of VIGS, the CWMV vector is more effective compared to that of TRV-VIGS and can be used for the silencing of miRNAs [126]. Effective gene silencing in greenhouses can also be accomplished by maintaining a cooler temperature constantly during the entire growth period of plants [136]. Furthermore, it has been found that lower inoculation temperatures improve *Agrobacterium* transformation efficiency [137]. At the same time, a certain humidity level is also important for the effectiveness of the silencing process [111]. Fu et al. (2016) discovered that at a lower humidity (30–40%), over 90% of tomato plants exhibited a silenced phenotype, and TRV was also efficient in infiltrating into flowers and fruits [138].

## 7. Future Perspectives

Even though VIGS is already a popular technique for analyzing gene function in non-model plants and accelerating research in model plants as well, several more recent advances will increase VIGS’s potential even more, for example, the investigations on heritable and long-term VIGS for crop enhancement have yet to be extended to further generations and additional genes (Figure 4).

Overall, developments in VIGS technology have quickened the finding of candidate genes involved in a variety of areas of plant biology, such as plant growth and development [139]. Some more recent advances will increase VIGS’s potential even more, such advances including transgenerational polyploidization, speciation, and epigenomics regulations in future plant breeding studies. The genes’ underlying crucial agronomic features are currently being discovered by combining VIGS with genome-wide association studies (GWASs) [140]. For the effective analysis of VIGS and VIGS-derived phenotyping, automated phenotyping platforms can be used [141].

The efficient VIGS system based on proper viral vector construction, the inoculation method, accurate positive controls, the best practices of plant care and of the maintenance of plant vigor, which can synergistically improve genomic and epigenomic regulatory mechanisms, will lead to the accurate annotation of the genome and the development of non-transgenics with a uniform phenotype at lower costs [132,142]. There are many economically important plant species, such as *Ananas comosus* (pineapple), *Theobroma cacao* (cocoa tree), and *Coffea canephora* (robusta coffee), and plants like *Amborella trichopoda* and *Arabidopsis lyrata* (Sand Cress), whose genomes have already been sequenced, however, there are no appropriate VIGS protocols for them [143,144,145].

## 8. Conclusions

VIGS is an efficient functional genomics tool for revealing modern plant biology. However, the recent developments in VIGS regarding the ability to induce heritable epigenetic modifications play a pivotal role in the gene’s identification, breeding, and production of genotypes with desirable traits. The ability to create modified plants that do not have a transgene but have altered gene expression and phenotypic traits will aid the regulatory approval of the release of genetically modified plants. However, the drawback of this approach, especially for breeding purposes, is that its impact is only momentary and is rarely transmitted to stable DNA methylation through RdDM, also including the potential for unintended consequences, such as the development of viral resistance. Nonetheless, it has recently been hypothesized that high-pressure dsRNA spraying directed towards the nucleus can result in stable RdDM [57,146]. Currently, VIGS-based technologies have broadened the genetic toolkit beyond gene silencing to include virus-induced genome editing (Table 1), overexpression, and host-induced gene silencing, expanding this toolbox for non-model species.

## Figures and Tables

**Figure 1 ijms-24-05608-f001:**
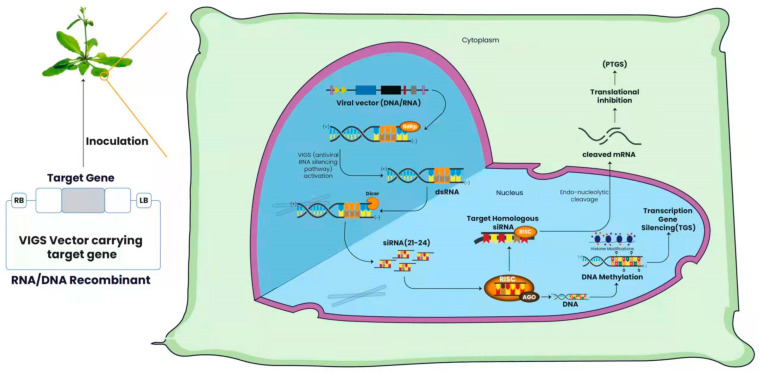
Molecular mechanisms of VIGS viral vectors (DNA or RNA) injected into plants containing the sequence of the intended gene. VIGS activates when the endogenous RDRP generates viral dsRNA, which are cleaved into small (21–24 nt) siRNAs by Dicer. Then, amplified siRNAs join with the AGO protein to form RISC, these siRNAs being used by RISC to precisely connect with homologous RNA in the cell, triggering endo-nucleolytic cleavage and translational inhibition of the cognate target mRNA, resulting in PTGS. The AGO complex also interacts with target DNA molecules in the nucleus, triggering transcriptional repression and TGS by DNA methylation and histone modifications. RNA-directed RNA polymerase (RDRP), transcriptional gene silencing (TGS), RNA Induced Silencing Complex (RISC), and short interference RNAs (siRNAs).

**Figure 2 ijms-24-05608-f002:**
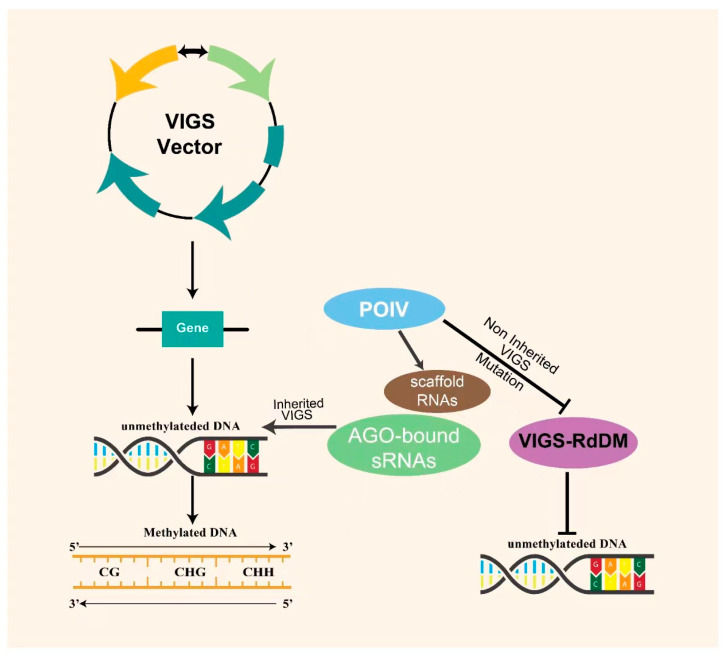
Establishment of epigenetic silencing when the VIGS vector is inserted into the cell of the targeted gene. PolV-synthesized scaffold RNAs provide binding sites to AGO sRNAs which direct heritable DNA methylation to induce gene silencing. A mutation in PolV results in the complete loss of VIGS-RdDM and non-heritable epigenetic silencing. ARGONAUTE (AGO), RNA-directed DNA Methylation (RdDM), Polymerase V (PolV).

**Figure 3 ijms-24-05608-f003:**
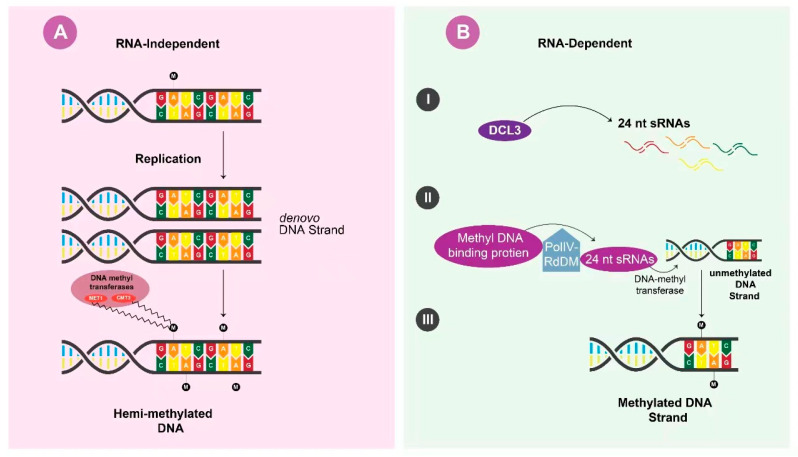
Maintenance of epigenetic silencing. (**A**) RNA-independent maintenance depends on the DNA methyltransferases MET1 and CMT3 recognizing hemimethylated Cs in the symmetrical context of de novo DNA strands for maintenance of epigenetic marks. (**B**) DCL3 synthesized 24nt sRNAs are recruited by PolIV-RdDM and loaded on the primary genomic location of RdDM by methyl DNA-binding proteins, and then DNA methyltransferase methylates the unmethylated strand to create gene silencing. PolIV-RdDM (Polymerase IV- RNA-directed DNA methylation).

**Figure 4 ijms-24-05608-f004:**
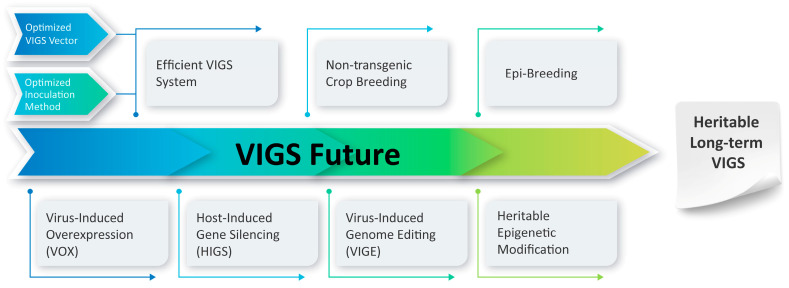
Future Direction of VIGS.

**Table 1 ijms-24-05608-t001:** Viral-Vector Used for Heritable Genome Editing.

Viral-Vector	Targeted-Gene	Next-Generation Efficiency	Delivery Method	Reference
Cotton leaf crumple virus (CLCrV)	*BRI1*, *GL2*, *PDS*	4.35–8.79%	*Agrobacterium*-mediated transient transformation	[64]
Tobacco rattle virus (TRV)	*AtPDS3*	30–60%	*Agrobacterium*-based flooding method	[65]
Potato virus X PVX	*NbXT2B*, *NbPDS3*, *NbFT*	100% for *NbXT2B* and 20% and 30% for *NbPDS3* and *NbFT*	Agroinfiltration	[68]
Pea early-browning virus PEBV	*PDS*	57 to 63%	*Agrobacterium* transformation	[69]
Beet necrotic yellow vein virus (BNYVV)	*NbPDS*	85%	*Agrobacterium*-mediated transformation method	[70]
Barley stripe mosaic virus (BSMV)	*TaPDS*, *TaGASR7*, and *TaGW2*	12.9% to 100%	*Agrobacterium*-mediated gene delivery	[67]

## Data Availability

Not applicable.

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
