# Peer review of "Virus-Induced Gene Silencing (VIGS): A Powerful Tool for Crop Improvement and Its Advancement towards Epigenetics"

_ijms, 2023, doi:10.3390/ijms24065608_

Round 1

Reviewer 1 Report

The manuscript entitled "Virus-Induced Gene Silencing (VIGS): A Powerful Tool for Crop Improvement and its Advancement Towards Epigenetics" depicts the role of VIGS in improving crop growth and development. I recommend major revision at this level. Highlight the novelty of this manuscript. In addition, please incorporate the below-mentioned comments.

General comments:

The authors in this manuscript reviewed a significant amount of literature on VIGS for improving crop growth. As a review, it does an excellent job of pointing out past findings and current understandings. However, this review needs to be placed in the context of previous reviews for a long-standing contribution, given the extensive interest in this topic. Clear statements about the content (and relevance?) of previous reviews need to be made to demonstrate how this review adds additional value to the previous reviews. The authors should provide significant new syntheses and insights at the end of each section rather than summarizing the literature. Improve the captions of tables and figures also.

Specific comments:

To further improve the text, I suggest the following changes in the manuscript.

1.     Please pay attention to the use of full stop and commas

2.     The review is quite interesting; however, I suggest improving the structure of the manuscript and English. Indeed, throughout the manuscript, several sentences don't make sense, appaired incomplete or are not bound to the previous or the following one. Please pay attention to punctuation and text formatting.

3.     Abstract is not well written; it is only a mere conscript of the study. Better would be to give some introduction followed by the gap in knowledge, hypothesis, general results and then a conclusion. The abstract is the only part of the paper that most readers see. Therefore, it is critically important for authors to ensure that their enthusiasm or bias does not mislead the reader. Please underscore the scientific value added to your paper in your abstract. Your abstract should clearly state the essence of the problem you are addressing, what you did and what you found and recommend. That will help prospective readers of the abstract to decide if they wish to read the entire article.

4.     Introduction section is not well presented and not strongly linked to the gaps in the research; therefore, the novelty of the work is not significant. Please improve the state-of-the-art overview to clearly show the progress beyond state-of-the-art. The lack of proper justification creates the wrong impression that the authors are unaware of the recent developments. A high-quality review paper has to provide a proper state-of-the-art analysis after the literature review and only based on the analysis to formulate the paper's goals. In addition, the introduction should be clearly stated the research questions and targets first. Then answer several questions: Why is the topic important (or why do you study it)? What are the research questions? What’s the gap in knowledge? Which is the scope of the manuscript? What hypothesis has been made? What has been studied? What are your contributions? The major defect of this study is the debate or argument is not clearly stated in the introduction session. At the end of the introduction, the statement of the paper's goal and the explanation of novelty has to be properly formulated. Currently, this is not performed well. The aim of the introduction should improve.

5.     Please add one section in which authors elaborate on how Virus-Induced Gene Silencing (VIGS) can be considered a powerful tool to improve plants growth which is grown under mitigate abiotic (salinity, heat, heavy metals, drought, etc.) and biotic (pathogens attack, diseases, etc.) stresses.

6.     Better to add future strategies and recommendation sections separately. 

7.     Better to add conclusions section separately; the conclusions can still be improved by providing an analysis of where the current work on adsorbents is focused and the remaining gaps in the literature where more research should be conducted. It is recommended to use quantitative reasoning compared with appropriate benchmarks, especially those stemming from previous work. Limitations in the suggested approach should be discussed in the conclusions section. Please add future work as well.

Author Response

Reviewer: 1

Responses to the editor and reviewers' comments

Dear Editor,

We are thankful to you for the critical review and constructive comments. The manuscript has been revised accordingly. The revision in the MS is marked up using the “Track Changes”. Hopefully, the revised version and accompanying response of our manuscript (ID: ijms-2177484 ) " Virus-Induced Gene Silencing (VIGS): A Powerful Tool for Crop Improvement and its Advancement Towards Epigenetics." will be sufficient to make the manuscript suitable for publication.

Thank you again for your valuable support. 

Dr. Aixia Gu

E-mail address: [email protected]

State Key Laboratory of North China Crop Improvement and Regulation,

Hebei Agricultural University, 071000 Baoding, China.

Response to Reviewer 1

We appreciate the assistant editor’s and referees’ comments; the suggestions of reviewers have helped us to increase the quality and content of the review. With the help of all the senior authors and group discussions, we have tried to address all the concerns raised in the revised version.

Please note the following changes in the revised version.

The manuscript entitled "Virus-Induced Gene Silencing (VIGS): A Powerful Tool for Crop Improvement and its Advancement Towards Epigenetics" depicts the role of VIGS in improving crop growth and development. I recommend major revision at this level. Highlight the novelty of this manuscript. In addition, please incorporate the below-mentioned comments.

General comments:

The authors in this manuscript reviewed a significant amount of literature on VIGS for improving crop growth. As a review, it does an excellent job of pointing out past findings and current understandings. However, this review needs to be placed in the context of previous reviews for a long-standing contribution, given the extensive interest in this topic. Clear statements about the content (and relevance?) of previous reviews need to be made to demonstrate how this review adds additional value to the previous reviews. The authors should provide significant new syntheses and insights at the end of each section rather than summarizing the literature. Improve the captions of tables and figures also.

Specific comments:

Response: First of all, we would like to express our deep gratitude for the invaluable comments that helped us revise our manuscript. We are in the process of addressing, revising, and submitting our manuscript. All of the concerns and suggestions you raised have now been addressed. The following will be our point-by-point responses.

To further improve the text, I suggest the following changes in the manuscript.

1.Please pay attention to the use of full stop and commas

Response: Dear reviewer Thank you for highlighting this the comment has addressed.

2.The review is quite interesting; however, I suggest improving the structure of the manuscript and English. Indeed, throughout the manuscript, several sentences don't make sense, appaired incomplete or are not bound to the previous or the following one. Please pay attention to punctuation and text formatting.

Response: Dear reviewer Thank you for appreciating our work. We have proofread the manuscript thoroughly and made required corrections in revised manuscript.

  1. Abstractis not well written; it is only a mere conscript of the study. Better would be to give some introduction followed by the gap in knowledge, hypothesis, general results and then a conclusion. The abstract is the only part of the paper that most readers see. Therefore, it is critically important for authors to ensure that their enthusiasm or bias does not mislead the reader. Please underscore the scientific value added to your paper in your abstract. Your abstract should clearly state the essence of the problem you are addressing, what you did and what you found and recommend. That will help prospective readers of the abstract to decide if they wish to read the entire article.

Response: Dear reviewer Thank you for your valuable suggestion. The comment has Addressed.

4.Introduction section is not well presented and not strongly linked to the gaps in the research; therefore, the novelty of the work is not significant. Please improve the state-of-the-art overview to clearly show the progress beyond state-of-the-art. The lack of proper justification creates the wrong impression that the authors are unaware of the recent developments. A high-quality review paper has to provide a proper state-of-the-art analysis after the literature review and only based on the analysis to formulate the paper's goals. In addition, the introduction should be clearly stated the research questions and targets first. Then answer several questions: Why is the topic important (or why do you study it)? What are the research questions? What’s the gap in knowledge? Which is the scope of the manuscript? What hypothesis has been made? What has been studied? What are your contributions? The major defect of this study is the debate or argument is not clearly stated in the introduction session. At the end of the introduction, the statement of the paper's goal and the explanation of novelty has to be properly formulated. Currently, this is not performed well. The aim of the introduction should improve.

Response: Thank you for raising this point. The introduction has been modified according to the suggested suggestions.

  1. Please add one section in which authors elaborate on how Virus-Induced Gene Silencing (VIGS) can be considered a powerful tool to improve plants growth which is grown under mitigate abiotic (salinity, heat, heavy metals, drought, etc.) and biotic (pathogens attack, diseases, etc.) stresses.

Response: Dear reviewer the comment has addressed with the heading name  ‘’Biotic and Abiotic Stress Tolerance and Their Potential for Crops Improvement via VIGS.’’ L342-381

  1. Better to add future strategies and recommendation sections separately.

Response: Dear reviewer the comment has addressed. Thank you for mentioning this. For future perspectives L473-497.

  1. Better to add conclusions section separately; theconclusions can still be improved by providing an analysis of where the current work on adsorbents is focused and the remaining gaps in the literature where more research should be conducted. It is recommended to use quantitative reasoning compared with appropriate benchmarks, especially those stemming from previous work. Limitations in the suggested approach should be discussed in the conclusions section. Please add future work as well.

Response: Dear reviewer Thank you for your valuable suggestion. We tried to address the comment according to the suggestions. L499-512

Reviewer 2 Report

This manuscript is well presented and up to date.However a few sentences appear unfinished to me.This applies to the sentence beginning at line 74 and the one at 145.The sentence starting at line 42 should be reformulated deleting"which favors the fittest and is subject to natural selection". In line 79 what is the meaning of" at chromatin locus" . 

At line 102"sustained"?

At line225 "the positive-strand RNA should be" the positive single-strand RNA".

The sentence starting at line 234 is not clear and should be revised.

Line 317:non-compliant? maybe compatible?

Author Response

Reviewer: 2

Responses to the editor and reviewers' comments

Dear Editor,

We are thankful to you for the critical review and constructive comments. The manuscript has been revised accordingly. The revision in the MS is marked up using the “Track Changes”. Hopefully, the revised version and accompanying response of our manuscript (ID: ijms-2177484 ) " Virus-Induced Gene Silencing (VIGS): A Powerful Tool for Crop Improvement and its Advancement Towards Epigenetics." will be sufficient to make the manuscript suitable for publication.

Thank you again for your valuable support. 

Dr. Aixia Gu

E-mail address: [email protected]

State Key Laboratory of North China Crop Improvement and Regulation,

Hebei Agricultural University, 071000 Baoding, China.

Response to Reviewer 2

We appreciate the assistant editor’s and referees’ comments; the suggestions of reviewers have helped us to increase the quality and content of the review. With the help of all the senior authors and group discussions, we have tried to address all the concerns raised in the revised version.

Please note the following changes in the revised version.

This manuscript is well presented and up to date. However, a few sentences appear unfinished to me. This applies to the sentence beginning at line 74 and the one at 145. ".

Response: Dear reviewer Thank you for mentioning this. The comment has addressed. L84 and L170.

In line 79 what is the meaning of" at chromatin locus" . 

Response: Dear reviewer A chromatin locus has a complex composition

‘The DNA sequence and the cell type determine the local chromatin composition and function. However, structural elements such as histones are ubiquitous chromatin components, and thus their presence, although biologically crucial, does not define a specific function’ [1,2]

The sentence starting at line 42 should be reformulated deleting" which favors the fittest and is subject to natural selection.

Response: Dear reviewer the comment has addressed, L51.

At line 102"sustained"?

Response: Dear reviewer the comment has addressed, L122.

At line225 "the positive-strand RNA should be" the positive single-strand RNA".

Response: Dear reviewer thank you for mentioning this the correction has made its " the positive single-strand RNA". L243

The sentence starting at line 234 is not clear and should be revised.

Response: Dear reviewer the comment has addressed, L251-252.

Line 317:non-compliant? maybe compatible?

Response: Dear Viewer, the correction has made its non-compatible, L402.

Reference

[1]         D.C. Baulcombe, C.J.C.S.H.p.i.b. Dean, Epigenetic regulation in plant responses to the environment, 6(9) (2014) a019471.

[2]         M. Vermeulen, J.J.N.R.M.C.B. Déjardin, Locus-specific chromatin isolation, 21(5) (2020) 249-250.

Reviewer 3 Report

This review discusses molecular mechanisms and experimental procedure of VIGS technique, and its applications in heritable epigenome modifications, gene function analysis and genetic editing. Although the structure of the content is appropriate, some sections have not fully developed while some paragraphs and sentences are redundant. In addition, language also needs improvement since some sentences are hard to read.

1.     In the introduction part, the explanation of the PTGS and TGS is put at the end paragraph after mentioning it at former paragraph, make the whole part quite redundant, I suggest making the mechanism discussion more concise and clearer.

2.     Add reference for the statement in line 78 “DNA methylation is a prerequisite for Pol V recruitment”.

3.     The content for section 2 about heritable epigenetics modification is not fully developed. It is good to explain the function of PolV and 3 DNA methylation pathway, but I feel the last paragraph about researches on inducing inheritable DNA methylation is not fully developed. Can you give more discussion on how the new improvement can guarantee the transgenerational epigenetic gene modification, and its application potential in other angiosperms?

4.     The sentence from line 172 to 176 is too long to read, please make it more simple and concise.

5.     Line 225 to 233 is about ssRNA formation, why it suddenly appears between content about application of VIGS vectors, it is something should be explained at the very beginning of the vector induced RNA interference part. Similarly, the paragraph from line 324 to 331 is also quite confusing. What is the link between hairpin structure of RNA and many species without appropriate VIGS protocols? Some more explanation is needed.

Author Response

Reviewer: 3

Responses to the editor and reviewers' comments

Dear Editor,

We are thankful to you for the critical review and constructive comments. The manuscript has been revised accordingly. The revision in the MS is marked up using the “Track Changes”. Hopefully, the revised version and accompanying response of our manuscript (ID: ijms-2177484 ) " Virus-Induced Gene Silencing (VIGS): A Powerful Tool for Crop Improvement and its Advancement Towards Epigenetics." will be sufficient to make the manuscript suitable for publication.

Thank you again for your valuable support. 

Dr. Aixia Gu

E-mail address: [email protected]

State Key Laboratory of North China Crop Improvement and Regulation,

Hebei Agricultural University, 071000 Baoding, China.

Response to Reviewer 3

We appreciate the assistant editor’s and referees’ comments; the suggestions of reviewers have helped us to increase the quality and content of the review. With the help of all the senior authors and group discussions, we have tried to address all the concerns raised in the revised version.

Please note the following changes in the revised version.

This review discusses molecular mechanisms and experimental procedure of VIGS technique, and its applications in heritable epigenome modifications, gene function analysis and genetic editing. Although the structure of the content is appropriate, some sections have not fully developed while some paragraphs and sentences are redundant. In addition, language also needs improvement since some sentences are hard to read.

  1. In the introduction part, the explanation of the PTGS and TGS is put at the end paragraph after mentioning it at former paragraph, make the whole part quite redundant, I suggest making the mechanism discussion more concise and clearer.

Response: Dear reviewer Thank you for highlighting this. The comment has addressed.

  1. Add reference for the statement in line 78 “DNA methylation is a prerequisite for Pol V recruitment”.

Response: Dear reviewer Thank you for mentioning it. The comment has addressed, L92.

  1. The content for section 2 about heritable epigenetics modification is not fully developed. It is good to explain the function of PolV and 3 DNA methylation pathway, but I feel the last paragraph about research on inducing inheritable DNA methylation is not fully developed. Can you give more discussion on how the new improvement can guarantee the transgenerational epigenetic gene modification, and its application potential in other angiosperms?

Response: Dear reviewer the function of PolV (L126-131) and 3 DNA methylation pathway (L123-126) has added. The improvements regarding transgenerational epigenetics gene modification and its application and its application potential in other angiosperms has added L182-188.

  1. The sentence from line 172 to 176 is too long to read, please make it simpler and more concise.

ResponseDear reviewer Thank you for highlighting it. The correction has made. L204-207

  1. Line 225 to 233 is about ssRNA formation, why it suddenly appears between content about application of VIGS vectors, it is something should be explained at the very beginning of the vector induced RNA interference part.

Response: Dear reviewer it is the molecular mechanism of RNA-based VIGS vector the correction has made L243-254.

Similarly, the paragraph from line 324 to 331 is also quite confusing. What is the link between hairpin structure of RNA and many species without appropriate VIGS protocols? Some more explanation is needed.

Response: Dear reviewer Thank you for your valuable questions the confusion has removed, and correction has made.

Round 2

Reviewer 1 Report

I still see places where the manuscript needs your attention

1. Please add the main objectives of this review in the last paragraph of the introduction section

2. Mention the appropriate reference where the figures are modified.  

Reviewer 3 Report

The suggestions have been appropriately addressed; I am good with the current version.